# JAK Signaling Is Critically Important in Cytokine-Induced Viral Susceptibility of Keratinocytes

**DOI:** 10.3390/ijms24119243

**Published:** 2023-05-25

**Authors:** Kimberly A. Arnold, Liam F. Peterson, Lisa A. Beck, Matthew G. Brewer

**Affiliations:** 1Departments of Dermatology, University of Rochester Medical Center, Rochester, NY 14642, USAlisa_beck@urmc.rochester.edu (L.A.B.); 2Pathology and Laboratory Medicine, University of Rochester Medical Center, Rochester, NY 14642, USA

**Keywords:** immune profiles, cytokines, Janus kinase inhibitors, keratinocyte, viral infection

## Abstract

Little is known about whether type 1 (IFNγ), 2 (IL-4/IL-13), or 3 (IL-17A/IL-22) cytokines affect the susceptibility of keratinocytes (KC) to viruses. These immune pathways predominate in various skin diseases: lupus, atopic dermatitis (AD), and psoriasis, respectively. Janus kinase inhibitors (JAKi) are approved to treat both AD and psoriasis, and are in clinical development for lupus. We evaluated whether these cytokines alter viral susceptibility of KC and determined if this effect is modulated by treatment with JAKi. Viral susceptibility to vaccinia virus (VV) or herpes simplex virus-1 (HSV-1) ± JAKi was assessed in immortalized and primary human KC pretreated with cytokines. Exposure to type 2 (IL-4 + IL-13) or the type 3 (IL-22) cytokines significantly increased KC viral susceptibility. Specifically, there was a peak increase of 12.2 ± 3.1-fold (IL-4 + IL-13) or 7.7 ± 2.8-fold (IL-22) in VV infection as measured by plaque number. Conversely, IFNγ significantly reduced susceptibility to VV (63.1 ± 64.4-fold). The IL-4 + IL-13-induced viral susceptibility was reduced (44 ± 16%) by JAK1 inhibition, while the IL-22-enhanced viral susceptibility was diminished (76 ± 19%) by TYK2 inhibition. IFNγ-mediated resistance to viral infection was reversed by JAK2 inhibition (366 ± 294% increase in infection). Cytokines expressed in AD skin (IL-4, IL-13, IL-22) increase KC viral susceptibility while IFNγ is protective. JAKi that target JAK1 or TYK2 reversed cytokine-enhanced viral susceptibility, while JAK2 inhibition reduced the protective effects of IFNγ.

## 1. Introduction

The immune system utilizes different cytokine profiles to respond to distinct pathogens. Typically, type 1, 2, and 3 cytokine profiles provide protection against viral, helminthic, and bacterial/fungal infections, respectively [1]. The prototypical cytokine(s) include IFNγ (type 1), IL-4 and IL-13 (type 2), as well as IL-17A and IL-22 (type 3). While these cytokines are important in response to pathogens, persistent expression can contribute to the development of chronic cutaneous diseases. Lupus, atopic dermatitis (AD), and psoriasis are examples of such diseases, which are recognized for their type 1, 2, or 3 inflammatory profiles, respectively [2,3,4].

A unique aspect of AD is the heterogeneous clinical presentation, including diverse cutaneous cytokine profiles, which can include IFNγ, IL-4 + IL-13, IL-17A, and IL-22 [5,6,7,8]. Some patients with AD demonstrate an increased propensity for disseminated cutaneous viral infections, which differs from patients with other chronic inflammatory skin diseases [9]. A key example of this occurrence is the life-threatening complications known as eczema herpeticum and eczema vaccinatum, which can arise after exposure to herpes simplex virus-1 (HSV-1) or vaccinia virus (VV), respectively [10,11,12]. Further, patients with a history of AD can have more severe infections from the recently emerged monkeypox virus [13]. However, it is still unclear why a subset of AD patients suffer from frequent cutaneous viral infections. It is possible that specific combinations of cytokines present in AD skin acti on the cell that viruses first infect (i.e., the keratinocyte (KC))and lead to uncontrolled viral replication and ultimately cutaneous and even systemic dissemination.

Many cytokines signal through the Janus kinase (JAK)–signal transducer and activator of transcription (STAT) pathways. As a result, this is an attractive target to treat inflammatory diseases. Multiple JAK inhibitors (JAKi) are currently FDA-approved or in late-stage clinical development to treat various inflammatory diseases. For example, topical ruxolitinib (marketed as JAK1- and JAK2-selective) and oral preparations of upadacitinib and abrocitinib (both JAK1-selective) are FDA-approved to treat AD [14,15,16]. Additionally, oral deucravacitinib (TYK2-selective) was recently approved for the treatment of plaque psoriasis [17]. Topical ruxolitinib and oral deucravacitinib are also being tested for treatment of cutaneous lupus manifestations (NCT04908280 and NCT04857034). With the development of JAKi for a wide variety of human conditions, it is important to understand whether—and if so, how—JAKi affect KC biology.

JAKi can affect the signaling of a wide range of cytokines and growth factors, which may explain the challenging safety profiles associated with this class of therapeutics. For example, clinical trials studying first generation, nonselective JAKi reported an increased incidence of viral infection, most often resulting from members of the Herpesviridae family (zoster and/or simplex). JAKi that have been shown to increase the risk of viral infections include orally administered ruxolitinib, tofacitinib, or baricitinib [18]. In contrast, topical ruxolitinib and delgocitinib (approved only in Japan to treat AD pruritus) reported few cases of Herpesviridae-related adverse events, but studies were not powered to determine whether there was a difference from the placebo-treated group [19,20]. Oral administration of selective JAKi, including abrocitinib, upadacitinib, and deucravacitinib, have also demonstrated increases in Herpesviridae-related adverse events [14,21,22,23,24]. While systemic administration of JAKi may be more effective than topical administration at reducing the signs and symptoms of chronic skin diseases, likely due to their actions on circulating immune cells, it is notable that even topical administration can enhance risk for cutaneous viral infections. Given that patients with AD already have an increased likelihood of disseminated cutaneous viral infection, assessing how JAKi impact viral infection in skin cells would be informative for determining risk potential.

We have previously shown that the type 2 cytokines IL-4 and IL-13 enhance the susceptibility of KC to viral infection [25]. To understand whether this observation is unique to type 2 cytokines, we evaluated cytokines characteristic of other inflammatory immune pathways: IFNγ (type 1), IL-17A (type 3), and IL-22 (type 3). Additionally, we demonstrated that complete inhibition of JAK signaling reversed the increased viral susceptibility observed in IL-4 + IL-13-stimulated KC [25]. To clarify the importance of specific JAKs in viral susceptibly, we utilized JAKi of varying selectivity on cytokine-stimulated KC infected with either VV or HSV-1 (Table 1).

## 2. Results

### 2.1. Type 2 (IL-4 + IL-13) and Type 3 (IL-22) Cytokines Increase KC Viral Susceptibility

Different cytokine profiles have been detected in the skin of AD patients including high levels of type 1, type 2, and type 3 cytokines [5,6,7,8,9]. In recognition of the differences in expression, we conducted a dose response to assess the effects of IFNγ, IL-4 + IL-13, IL-17A, or IL-22 on the susceptibility of the immortalized KC cell line N/TERT-2G to VV infection (24–48 h, Figure 1). Since we recently identified that KC differentiation is a critical determinant of viral susceptibility [26], two different timelines of differentiation/infection were necessary to detect either decreases (24 h post differentiation) or increases (48 h post differentiation) in KC viral susceptibility (Appendix A).

After exposure to a low MOI (0.0001) of VV, infection was quantified by plaque number. A viral plaque represents a cleared area of the monolayer which arises because the infection has lysed cells. Another common measurement of infection is qPCR for viral genomes. However, plaque assays detect both viral spread and cytopathic effect, which is not possible with qPCR analysis. Treatment with IL-4 + IL-13 at all concentrations, as well as IL-22 at the highest tested concentrations (50 and 200 ng/mL), significantly increased plaque number (Figure 1A). IL-17A had no effect on KC susceptibility to VV infection. To assess whether the VV findings were seen with other cutaneous viruses, we next investigated susceptibility of KC under these conditions using HSV-1. Consistent with VV observations, KC treated with IL-4 + IL-13 (*p* < 0.05) or IL-22 (similar trend) and then infected with HSV-1 showed an increase in plaque number, while IL-17A had no effect (Appendix A). Taken together, this suggests that cytokines expressed in AD skin increase KC viral susceptibility and these cytokine-induced changes may be relevant for KC susceptibility to multiple cutaneous viral pathogens (Table 2).

### 2.2. The Type 1 Cytokine IFNγ Decreases KC Viral Susceptibility

Conversely, to determine reductions in viral susceptibility, infection must occur at the stage of differentiation where KC demonstrate elevated susceptibility. We have demonstrated that viral susceptibility is elevated acutely after induction of differentiation [26]. For this reason, infections with IFNγ treatment (Figure 1B) occurred at an earlier timepoint (differentiated for 24 h) than the results in Figure 1A (differentiated for 48 h), which explains the high number of plaques in the media control. While untreated wells demonstrated robust infection as indicated by >200 plaques per well, all doses of IFNγ significantly decreased plaque number (Figure 1B). These results were recapitulated with HSV-1 infection (Appendix A).

### 2.3. Neither Cytokine Nor JAKi Exposure Affects KC Viability

To ensure that alterations to cellular viability were not driving the differences observed in infection, viability of cytokine-treated cultures was determined. This was achieved using two different metrics, LDH (membrane integrity) and WST-1 (metabolic activity) assays. No cytokines demonstrated a significant change in KC viability through either assay (Figure 2A).

JAK inhibition has been successful in treatment of chronic inflammatory skin diseases [27,28,29]. This has led to FDA approval of topical ruxolitinib (marketed as JAK1- and JAK2-selective) and oral preparations of abrocitinib and upadacitinib (both JAK1-selective) for AD, as well as oral deucravacitinib (TYK2-selective) for plaque psoriasis. However, the effects of JAKi on KC viral susceptibility have not been studied. To determine whether JAKi treatment alters cytokine-induced KC viral susceptibility, we screened several JAKi with varying selectivity (Table 1). We first determined whether JAKi altered KC viability at a concentration comparable to what is observed in patients’ serum (Table 1). Neither LDH nor WST-1 assays showed any change from untreated KC, indicating that JAKi at these clinically relevant concentrations do not alter KC viability (Figure 2B). Plaque assays were then used to assess whether JAKi alter KC susceptibility to viral infection. Treatment with JAKi alone did not affect the susceptibility of KC to VV infection (Appendix A), demonstrating that JAK inhibition by itself does not alter viral susceptibility.

### 2.4. JAK1 Inhibition Reduces Viral Susceptibility of IL-4 + IL-13-Treated KC

We next evaluated the contribution of different JAK pathways to the increase in viral susceptibility observed with IL-4 + IL-13 or IL-22 treatment of KC. KC were treated with these cytokines for 48 h of differentiation. Twenty-four hours into differentiation, KCs were exposed to JAKi. Changes to KC viral susceptibility were then assessed by plaque assay. There was a significant decrease in VV plaque number in IL-4 + IL-13-treated KC cultures exposed to pyridone 6 and abrocitinib, and a dose-dependent reduction was also observed with ruxolitinib (Figure 3).

### 2.5. TYK2 Inhibition Reduces Viral Susceptibility of IL-22-Treated KC

However, only deucravacitinib significantly reduced plaque number in a dose-dependent fashion in IL-22-treated KC (Figure 4).These results suggest cytokines utilize particular JAK pathway(s) that alter the KC response to viral infection. Specifically, IL-4 + IL-13 signaling through JAK1 and IL-22 signaling through TYK2 enhances viral susceptibility in the epidermis (Table 2). 

### 2.6. JAK2 Inhibition Reverses IFNγ-Induced Resistance to VV Infection

Since IFNγ treatment reduced viral susceptibility of KC to VV infection (Figure 1B), we investigated whether treatment with selective JAKi reversed this resistance. Concurrent treatment of KC with IFNγ and ruxolitinib or fedratinib significantly increased plaque number (Figure 5). Extending these observations to another cutaneous viral pathogen, we observed similar observations in HSV-1-infected KC (Appendix A). Overall, these results suggest that IFNγ signaling through JAK2 diminishes KC viral susceptibility (Table 2).

### 2.7. A JAK1/JAK2-Selective Inhibitor (Ruxolitinib) Increases Virion Production from IFNγ-Treated KC

Since plaque formation can be influenced by the amount of virus produced from an infected cell, we wanted to understand whether JAK signaling after cytokine treatment impacts production of viral progeny. To do this, the amount of infectious VV released 24 h post infection was quantified in IL-4 + IL-13-, IL-22-, or IFNγ-treated KC ± JAKi. We observed a significant increase in the production of infectious VV in cultures treated with IL-4 + IL-13 or IL-22, but JAKi exposure did not reverse this increase (Appendix A). Conversely, IFNγ treatment alone significantly decreased the amount of infectious virus recovered from KC 24 h post infection. In this condition, JAKi did reverse the IFNγ-mediated decrease, with ruxolitinib significantly increasing viral titers. Other JAKi showed a reproducible trend of reversing IFNγ-induced viral susceptibility (pyridone 6, fedratinib, and deucravacitinib), but due to the high variability in viral titers observed from this model, statistical significance was not achieved (Appendix A).

### 2.8. Primary KC Substantiates the Importance of Cytokine Profiles and JAKi in Cutaneous Viral Susceptibility

We recapitulated key findings in primary human foreskin KC (PHFK). Since we had previously observed that IL-4 + IL-13 treatment enhances viral susceptibility of KC [25], we focused on the novel findings regarding IL-22 and IFNγ. Both IL-22 and IFNγ treatment of PHFK demonstrated a similar phenotype, with IL-22 significantly enhancing and IFNγ significantly decreasing KC susceptibility against VV infection (Figure 6A,B).

In line with the JAKi findings from immortalized KC, PHFK treated with ruxolitinib significantly reversed, and fedratinib showed a reproducible trend in reversing, IFNγ-mediated resistance to viral infection (Figure 6C). These results confirm that the cytokine profile present in the skin is critically important for KC viral susceptibility and that JAK inhibition can reverse cytokine-mediated effects (Table 2).

## 3. Discussion

Previous observations by our group identified that KC have increased susceptibility to viral infection with VV after exposure to type 2 cytokines (IL-4 + IL-13) [25]. We determined susceptibility of KC to VV and HSV-1 infection after exposure to representative cytokines prevalent in diverse inflammatory cutaneous diseases: IFNγ (lupus/AD; type 1), IL-4 and IL-13 (AD; type 2), IL-22 and IL-17A (psoriasis/AD, type 3). Our publication also indicated that the enhanced susceptibility to VV infection observed in KC treated with IL-4 + IL-13 could be reduced by exposure to the pan JAKi pyridone 6 [25]. To understand the importance of specific JAK signaling, KC were treated with cytokines ± JAKi of varying selectivity to determine changes in viral susceptibility.

The type 1 cytokine IFNγ is known to play a vital role in the antiviral response. This has been supported in mouse models of infection with Ebola, West Nile, and murine corona viruses, where IFNγ was critical in controlling these pathogens [30,31,32]. It has also been shown that a global knock-out of IFNγ in mice enhances susceptibility to VV infection [33]. Furthermore, studies have demonstrated diminished viral susceptibility to HSV-1 in primary KC treated with IFNγ [34,35]. Our studies are in line with these observations and demonstrate that IFNγ treatment reduces susceptibility of KC to both HSV-1 and VV infection. Our findings are novel in that they extend these observations of IFNγ-mediated protection from infection in KC to a poxvirus (VV) and investigate viral susceptibility in the context of JAKi exposure. IFNγ is known to signal in part through the JAK/STAT pathway, with evidence that JAK2 and STAT1 are critical [36,37]. Impaired STAT1 signaling has been shown to increase susceptibility to monkeypox infection in a murine model [38]. Human clinical trials with fedratinib (JAK2-selective) have not reported any virus-related adverse effects [39,40]. This could be explained by the fact that the drug reaches micromolar concentrations in the blood (Cmax 3.4 µM, Table 1), which would mean it is no longer JAK2-selective. Therefore, our research highlights the possibility that a highly JAK2-selective inhibitor would enhance the risk for a cutaneous viral infection. As new JAKi are developed, clinical trials enrolling populations that are at increased risk for cutaneous viral infections, such as AD patients, may require close monitoring for herpes simplex, varicella zoster, or molluscum contagiosum infections. This would be of greatest concern when the JAKi is highly selective for JAK2 at the concentrations achieved in vivo. Given that these findings were made in KC models independent of immune cells, an important future direction is to recapitulate these observations in vivo using transgenic mouse models where individual and combinations of cytokines are overexpressed using an epidermal promoter. This would further substantiate the importance of JAK2 signaling in the skin as a protective response to viral infection.

Several groups have demonstrated that KC exposed to the type 2 cytokines IL-4 and IL-13 have increased susceptibility to viral infection [41,42]. The results presented in this study reconfirm this observation in both primary and immortalized KCs and provide a more mechanistic understanding that this occurrence is driven by JAK1 signaling, as indicated by a reduction in viral susceptibility after JAK1 inhibition (abrocitinib). It is worth noting that the level of JAKi present in the skin following systemic administration of FDA-approved JAKi (abrocitinib, upadacitinib, fedratinib, and deucravacitinib) has to our knowledge never been measured. A caveat of our results is that systemic use of abrocitinib has shown a modest but reproducible association with virus-related adverse events [21,22,23]. We hypothesize that this is explained by abrocitinib achieving Cmax values as high as 2.1 µM following delivery of 100 mg QD (Table 1), which would expand the selectivity to JAK1, JAK2, and TYK2. Topical abrocitinib use has not yet been tested in clinical trials, but we predict from our findings that it may benefit subjects that exhibit a type 2 cytokine skin environment and suffer from recurrent cutaneous infections.

Similar to IL-4 + IL-13 treatment, the type 3 cytokine IL-22 also enhanced viral susceptibility, while treatment with IL-17A, another type 3 cytokine, had no effect on viral susceptibility. The effect of IL-22 on KC susceptibility to viral infection is incompletely studied. Anecdotal evidence observed in a single clinical case study showed that more severe HPV infection (as measured by wart count) positively correlated with IL-22 levels in the serum, supporting our observation that this cytokine promotes cutaneous viral susceptibility [43]. Our findings support the hypothesis that unique cytokine signatures present in the skin, e.g., subjects with the greatest expression of IL-4, IL-13, and IL-22, would be at most risk of experiencing disseminated viral infections. IL-22 has been linked to keratinocyte proliferation and prevention of differentiation [44,45]. Our previous finding that differentiation contributes to KC susceptibility to viral infection may explain why this occurs [25,46]. From these two observations, we hypothesize that IL-22 alters the differentiation kinetics of KC and may keep KC in a state highly permissive to viral infection. This is a possible explanation of why we observed a more modest, but still significant, effect of IL-22 on primary KC compared to immortalized cells, because of donor-specific variability in KC differentiation kinetics. Importantly, we found that inhibition of TYK2 signaling in IL-22-treated KC reduced susceptibility to VV infection. A unique characteristic of deucravacitinib is its remarkable selectivity for TYK2 and a Cmax in subjects (~100 µM) that would maintain its TYK2 selectivity. Nevertheless, recent clinical trials of deucravacitinib for the treatment of plaque psoriasis have shown a modest increase in virus-related adverse events compared to placebo [24,47]. Why this occurs is unclear, but similar to other JAKi that are systemically delivered, we hypothesize it is mediated in part through an impaired adaptive immune response, which overrides any beneficial effects exerted at the tissue or epidermal level.

Combining our observations from both IL-4 + IL-13- or IL-22-treated KC suggests that a topical JAK1 and TYK2 inhibitor applied to AD skin may have the highest likelihood of reducing the development of disseminated cutaneous viral infections. Since JAKi are in development for the treatment of many diseases, and it has been reproducibly observed that treatment with some JAKi can result in increased risk of infection, it is imperative that we characterize the effects of JAKi on the skin. Specifically, understanding how JAKi alter KC, the cell initially infected with the virus, and the immune cells necessary for protection against viral infections is critical in ascertaining how these drugs influence overall viral susceptibility and disease severity. Our results indicate in two different viral models (VV and HSV-1) that JAK2-selective inhibitors reduce IFNγ-mediated protection from viral infection in KC. This suggests that JAK2 is important in the host response to viral infection, and that therapies which diminish this signaling pathway in the skin could enhance the likelihood of developing severe cutaneous viral infections. Since we have recently experienced two viral pandemics (SARS-CoV-2 and monkeypox virus), both of which have cutaneous manifestations, it is important to identify patient populations that have elevated likelihood of severe disease and methods to prevent or mitigate serious and possibly life-threatening complications [48,49]. In this regard, we hypothesize that the low rate (~3%) of AD patients who experience EH would have the highest levels of IL-4, IL-13, and IL-22 in their skin [50]. We can validate these findings in humans and move closer to a personalized medicine approach, possibly by topically delivering JAKi selective for JAK1/TYK2 to individuals at greatest risk for life-threatening viral diseases.

## 4. Materials and Methods

### 4.1. Cells and Culture Techniques

The immortalized human KC line N/TERT-2G and primary human foreskin KC (PHFK) were propagated as previously described [46,51]. Briefly, N/TERT-2Gs were maintained in keratinocyte serum free media (KSFM) 1X (Invitrogen/Gibco, Grand Island, NY, USA) supplemented with bovine pituitary extract and epidermal growth factor (Invitrogen/Gibco, Grand Island, NY, USA), penicillin/streptomycin (Invitrogen/Gibco, Grand Island, NY, USA), amphotericin B (Invitrogen/Gibco, Grand Island, NY, USA), and 0.3 mM CaCl_2_ (Boston Bioproducts, Ashland, MA, USA). Cells were grown to 30% confluency, trypsinized, and plated. Differentiation was initiated by exposing cells to high calcium-containing (1.8 mM) Dulbecco’s modified Eagle medium (DMEM—Invitrogen/Gibco, Grand Island, NY, USA) [52]. Every two days, cells received fresh media. BSC40 cells were obtained from the ATCC and maintained in DMEM (Corning, Manassas, VA, USA) supplemented with 10% fetal bovine serum (Invitrogen/Gibco, Grand Island, NY, USA).

### 4.2. Cytokines and JAKi

IL-4, IL-13, IL-22, and IFNγ (Cat No.: 574004, 571104, 571304, 570204, respectively) were purchased from Biolegend (San Diego, CA, USA, and IL-17A (Cat No.: 317ILB) was purchased from R&D Systems (Minneapolis, MN, USA). These cytokines were used at concentrations ranging from 3.1 to 200 ng/mL. IL-4 and IL-13 were used at a 1:1 ratio in all experiments. The JAKi used in these experiments—ruxolitinib, abrocitinib, fedratinib, ritlecitinib, and deucravacitinib (Cat No.: HY-50856, HY-107429, HY-10409, HY-100754, HY-117287, respectively)were purchased from MedChemExpress (Monmouth Junction, NJ, USA), and pyridone 6 (Cat No. 420097) was purchased from Millipore Sigma (Saint Louis, MO, USA). All JAKi were used at concentrations ranging from 50 to 400 nM. The JAKi vehicle was dimethyl sulfoxide (DMSO) diluted to the highest concentration (0.4%) used in JAKi experiments.

### 4.3. Cell Viability Assays

The WST-1 metabolic assay was performed as previously published (36). Cells were exposed to IL-4 + IL-13, IL-22, or IFNγ at doses ranging from 3.1 to 200 ng/mL, or JAKi at 400 nM, and viability was assessed 24 or 48 h later. Absorbance readings (450 nm and 620 nm) were taken using a Spectramax i3x Multi-Mode Plate Reader (Molecular Devices, San Jose, CA, USA). Values were normalized to the percent of media control. LDH assays were performed by plating N/TERT-2G cells at a density of 75,000 cells per well in a 96-well plate. These cells were grown to confluency and then exposed to cytokines or JAKi treatments as noted above. Supernatants were collected from each well and the LDH assay was performed (Cytotoxicity Detection Kit^PLUS^ LDH, Millipore Sigma Saint Louis, MO, USA). The LDH detection reaction was conducted at room temperature and terminated by the addition of Stop Solution after 10 min. Readings (absorbance at 490 nm and 620 nm) were taken using a Spectramax i3x Multi-Mode Plate Reader (Molecular Devices, San Jose, CA, USA). The absorbance values from media-only wells were subtracted from each condition, and all values were normalized to a lysed control sample (representing 100% death) generated through addition of a 1:10 dilution of the provided lysis buffer.

### 4.4. Viral Plaque Assay

The Western Reserve (WR) strain of VV and the KOS strain of HSV-1 were used for all experiments [25,53]. Plaque assays were performed as previously described [46]. Cells were seeded in a 24-well plate (150,000 cells/well) and then grown to confluence. Cells were then switched to high calcium-containing media (1.8 mM) to induce differentiation. At the time of differentiation, KC were treated with IL-4 + IL-13, IL-22, IL-17A, or IFNγ and/or JAKi. An important note is that two different timelines of differentiation were used in our cytokine exposure experiments. We have previously shown that KC are at a state of elevated susceptibility to viral infection early after initiation of differentiation (0–24 h) [26]. Since IFNγ has been shown to reduce viral infection, we infected KC at 24 h post differentiation since cells were highly susceptible to infection. Conversely, to detect increases in viral susceptibility, we infected cells at a timepoint where they are resistant to infection (48 h). Cells treated with IL-4 + IL-13, IL-22, or IL-17A were differentiated for 48 h prior to viral infection. In certain conditions with IL-4 + IL-13 or IL-22 where cells were treated with both cytokines and JAKi, JAKi were added 24 h after cytokine treatment. Cells treated with IFNγ were differentiated for 24 h prior to viral infection. In this condition, IFNγ and JAKi were added together at the time of differentiation. After differentiation and treatment, cells were infected with a low multiplicity of infection ((MOI) 0.0001) of VV or HSV-1 for 48 or 72 h, respectively. The media was then removed and replaced with crystal violet solution for 2–3 h to stain for plaque visualization and quantitation. Monolayers were washed with water to remove crystal violet solution, and plates were allowed to dry completely prior to scanning and analysis with ImageJ (https://imagej.nih.gov/ij/download.html, accessed on 20 September 2017) 

### 4.5. Viral Titering

Cells were plated at a density of 150,000 cells per well in a 24-well plate. After confluency, cells were differentiated in the presence of either IL-4 + IL-13 or IL-22 at 50 ng/mL for 24 h, followed by addition of JAKi at 200 nM for 24 h or treated with IFNγ at 12.5 ng/mL and JAKi at 400 nM for 24 h. Treated cells were infected with a low multiplicity of infection ((MOI), 0.0001) of VV and then harvested one day later by scraping. Cells were freeze/thawed 3 times prior to titering and then sonicated for one minute to disaggregate virus. Infectious VV progeny were titered on BSC40 cells in 6-well plates (plated at 500,000 cells/well) as previously described [25]. Confluent monolayers were infected for 2 h with serially diluted cell lysate. Infection media was removed, cells were overlaid with DMEM containing 5% fetal bovine serum and methylcellulose, and then incubated at 37ºC for 2 days. BSC40 monolayers were then stained with crystal violet, and plaque number was enumerated to calculate viral titer in pfu/mL.

### 4.6. Statistical and Data Analysis

Statistical differences for all experiments were tested via a one-way ANOVA (paired when appropriate) with the Geisser–Greenhouse correction or a parametric paired ratio t-test. For cells treated with cytokine alone, no normalization was performed (Figure 1 and Figure 6, Appendix A). Plaque assays with both cytokines and JAKi were normalized to the vehicle control (DMSO) and cytokine combination (Figure 3, Figure 4, Figure 5 and Figure 6 and Appendix A). All statistical tests and graphs were performed with GraphPad Prism software v9.2.0 (GraphPad, San Diego, CA, USA). 

## Figures and Tables

**Figure 1 ijms-24-09243-f001:**
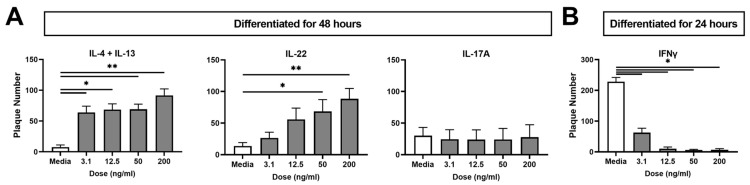
Cytokines Representative of Type 1, 2, and 3 Immunity Affect the Susceptibility of KC to Vaccinia Virus (VV) Infection. Immortalized KC (N/TERT-2G) were differentiated in calcium-containing media ± 3.1–200 ng/mL IL-4 + IL-13, IL-22, or IL-17A for 48 h (**A**), or IFNγ for 24 h (**B**) followed by a low multiplicity of infection ((MOI), 0.0001) with VV. Crystal violet staining was used to visualize plaques. IL-4 + IL-13 *n* = 5 and IL-22, IL-17A, IFNγ *n* = 4 experiments. Data are shown as mean ± SEM. * *p* < 0.05, ** *p* < 0.01.

**Figure 2 ijms-24-09243-f002:**
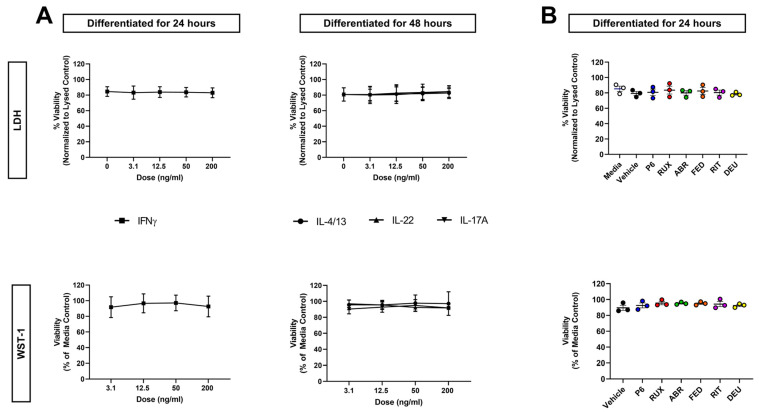
Neither Cytokine nor JAKi Treatment Affect Viability of KC. KC were differentiated for 24 h in the presence of IFNγ or 48 h with IL-4 + IL-13, IL-17A, or IL-22 (**A**). KC were exposed to Janus kinase inhibitors (JAKi) with varying selectivity at 400 nM (the highest concentration used in our studies) in differentiation media for 24 h (**B**). The vehicle was 0.4% DMSO. Viability was assessed by both the WST-1 and LDH assays. IL-4 + IL-13, IL-22, IFNγ *n* = 4 and IL-17A *n* = 3 experiments. JAKi *n* = 3 experiments. Data are shown as mean ± SEM.

**Figure 3 ijms-24-09243-f003:**
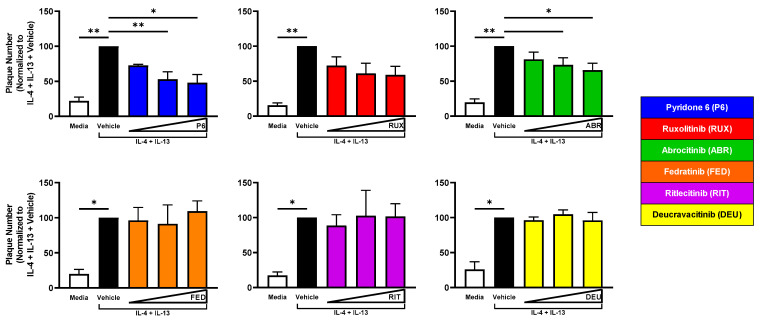
JAK1 Inhibition Reduces Susceptibility of IL-4 + IL-13-Treated KC to VV Infection. KC were differentiated ± 50 ng/mL of IL-4 + IL-13 for 24 h. Then, JAKi (50–200 nM) of varying selectivity were added to cells for 24 h. At 48 h post differentiation, cells were infected with a low MOI (0.0001) of VV. Crystal violet staining was used to visualize plaques. P6 *n* = 5; RUX, ABR *n* = 6; FED *n* = 4; RIT, DEU *n* = 3 experiments. Data are shown as mean ± SEM. * *p* < 0.05, ** *p* < 0.01.

**Figure 4 ijms-24-09243-f004:**
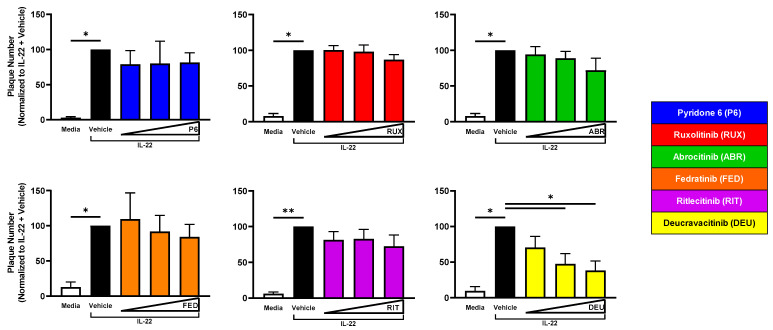
TYK2 Inhibition Reduces Susceptibility of IL-22-Treated KC to VV Infection. KC were differentiated ± 50 ng/mL IL-22 for 24 h. Then, JAKi (50–200 nM) of varying selectivity were added to cells for 24 h. At 48 h post differentiation, cells were infected with a low MOI (0.0001) of VV. Crystal violet staining was used to visualize plaques. P6 *n* = 3; RUX, ABR *n* = 4; FED, RIT, DEU *n* = 5 experiments. Data are shown as mean ± SEM. * *p* < 0.05, ** *p* < 0.01.

**Figure 5 ijms-24-09243-f005:**
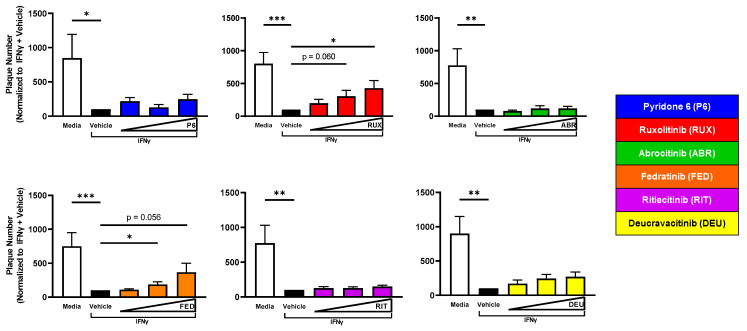
JAK2 Inhibition Reverses IFNγ-Induced Resistance to VV Infection in KC. KC were differentiated ± 12.5 ng/mL of IFNγ in tandem with JAKi (100–400 nM) of varying selectivity for 24 h. At 24 h post differentiation, cells were infected with a low MOI (0.0001) of VV. Crystal violet staining was used to visualize plaques. P6 *n* = 3; RUX *n* = 6; ABR *n* = 4; FED *n* = 5; RIT, DEU *n* = 4 experiments). Data are shown as mean ± SEM. * *p* < 0.05, ** *p* < 0.01, *** *p* < 0.001.

**Figure 6 ijms-24-09243-f006:**
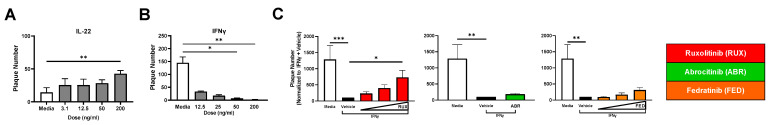
Primary Human Foreskin Keratinocytes (PHFK) Recapitulate Key Findings From Immortalized KC. PHFK were differentiated ± IL-22 for 48 h (**A**), 12.5 ng/mL of IFNγ alone (**B**) or in tandem with JAKi (100–400 nM) for 24 h (**C**). Cells were infected with a low MOI (0.0001) of VV. Crystal violet staining was used to visualize plaques. IL-22 *n* = 5; IFNγ *n* = 4; JAKi *n* = 3 donors. Data are shown as mean ± SEM. * *p* < 0.05, ** *p* < 0.01, *** *p* < 0.001.

**Table 1 ijms-24-09243-t001:** The Clinical Characteristics of the Janus Kinase Inhibitors (JAKi) Used in This Study. Inhibitory concentration of JAKi was determined by a Janus kinase activity assay. QD, once daily. BID, twice daily. * Tested against murine JAK; # topical treatment only.

Janus Kinase Inhibitor (Shorthand Nomenclature)	For Treatment of:	IC_50_ (nM)					Cmax		
		JAK1	JAK2	JAK3	TYK2	REFERENCE	nM	REFERENCE	Human Dose
**Pyridone 6 (P6)**	**Not in clinical** **development**	15 *	1	5	1	PMID: 11934592 Access Date: 15 July 2021	N/A	N/A	N/A
**Ruxolitinib (RUX)**	**Atopic dermatitis ^#^, Myelofibrosis, Polycythemia Vera, Graft-Versus-Host Disease**	3.3	2.8	428	19	PMID: 20130243 Access Date: 15 July 2021	650	PMID: 35368221 Access Date: 7 December 2022	15 mg BID
**Abrocitinib (ABR)**	**Atopic Dermatitis**	29.2	803	>10^4^	1250	https://www.accessdata.fda.gov/drugsatfda_docs/nda/2022/213871Orig1s000MultidisciplineR.pdf Access Date: 15 July 2021	2164	https://www.accessdata.fda.gov/drugsatfda_docs/nda/2022/213871Orig1s000MultidisciplineR.pdf Access Date: 7 December 2022	100 mg QD
**Fedratinib (FED)**	**Myelofibrosis**	105	3	1002	405	PMID: 18394554 Access Date: 15 July 2021	3438	https://packageinserts.bms.com/pi/pi_inrebic.pdf Access Date: 7 December 2022	400 mg QD
**Ritlecitinib (RIT)**	**Not FDA approved** **(In multiple clinical trials)**	>10^4^	>10^4^	33.1	>10^4^	PMID: 27791347 Access Date: 15 July 2021	N/A	N/A	N/A
**Deucravacitinib (DEU)**	**Plaque Psoriasis**	>10^4^	>10^4^	>10^4^	0.2	PMID: 31318208 Access Date: 15 July 2021	105	https://packageinserts.bms.com/pi/pi_sotyktu.pdf Access Date: 7 December 2022	6 mg QD

**Table 2 ijms-24-09243-t002:** Effect of Cytokines ± JAKi on KC Viral Susceptibility to VV Infection. # = Recapitulated in PHFK. & = Similar observations observed with HSV-1.

		**Effect on Viral Susceptibility**					
			**Ruxolitinib**	**Abrocitinib**	**Fedratinib**	**Ritlecitinib**	**Deucravacitinib**
			**Clinical Observations**				
		**No JAKi**	Observed (Significant)	Observed	Not Observed	No Reported Results	Observed
		**in vitro Observations (Vaccinia Virus)**					
**Cytokine**	IL-4 + IL-13	Increase ^&^	Decrease	Decrease	No effect	No effect	No effect
	IL-22	Increase ^#,&^	No effect	No effect	No effect	No effect	Decrease
	IL-17A	No effect ^&^	Not Determined				
	IFNγ	Decrease ^#,&^	Increase ^#,&^	No effect	Increase ^#,&^	No effect	No effect

## Data Availability

The data presented in this study are available on request from the corresponding author.

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
