# Peer review of "JAK Signaling Is Critically Important in Cytokine-Induced Viral Susceptibility of Keratinocytes"

_ijms, 2023, doi:10.3390/ijms24119243_

Round 1

Reviewer 1 Report

The author demonstrated cytokine-induced virus susceptibility via the Jak pathway using three different types of Jaki.

The following suggestion for improvements:

1.      The author should provide a comparison between the virus susceptibility induced by cytokine stimulation with IL-4 and IL-13 alone versus their combined effects as IL-4+IL-13.

2.      Additionally, the author should provide the respective addition ratio of IL-4 and IL-13 in the total amount of IL-4+IL-13.

3.      In section 4.1, Cells and Viruses, there is a lack of detailed procedures for virus infection.

Author Response

Included in the attachment is a point-by-point response to your suggestions.

Reviewer 2 Report

In this study, Arnold et al. evaluated the effects of different cytokine treatments and JAK inhibitors on keratinocyte viral susceptibility. The high points of the study are the use of multiple JAK inhibitors and the confirmation of results with two viruses. The authors have used plaque formation as a primary readout for the experiments shown in the manuscript. Following see my comments and suggestions,

1.     Why different timing for differentiation is used in this study? If the logic is that cells are more susceptible at 24h of differentiation, then 24h should be used for all the cytokine treatments.

2.     Figure 1A shows differences in plaque numbers between all the media controls for different cytokine treatments at 48h of differentiation. Is the strength of the virus varying through various experiments?

3.     Show keratinocyte differentiation markers as the control for differentiation (KRT1, KRT10, LOR, and FLG).

4.     How much of a difference would we expect for viral infection in two different states of keratinocytes- proliferation/differentiating? In other words, why was keratinocyte differentiation preferred over proliferation?

5.     Show positive controls for cytokine treatments. Preferably qRT-PCR for downstream pathway genes.

6.     Show positive controls for JAK inhibition, such as immunoblotting for STAT phosphorylation sites.

7.    Show macroscopic images of plaque formation in Figures 1, 3, 4, 5, and 6.

Author Response

(The authors gave the same response as above.)

Round 2

Reviewer 1 Report

After the initial round of review, there are still some issues that need to be clarified.

1.      Based on the first-round feedback, the reviewer would like to know if there is a dose-dependent effect of IL-4 and IL-13 stimulation on plaque generation.

2.      Additionally, was the use of 50 ng/ml of IL-4 and IL-13 the optimal dose for cytokine stimulation?

3.      Based on the provided figure, what is the significance of 50 ng/ml of IL-4+IL-13? Does it represent 25ng/ml of IL-4 and 25ng/ml of IL-13?

4.      What would be the result of cytokine stimulation when the optimal dose of IL-4 is combined with the optimal dose of IL-13?

Based on the study results mentioned above, it would be reasonable to select a 1:1 combination of IL-4 and IL-13 for cytokine stimulation. Otherwise, the reviewer recommended that the author conduct additional experiments to demonstrate the individual impact of IL-4 and IL-13 separately.

Minor:

1.      Please provide subtitles for each study result. This will help the reader better understand the purpose of this study.

Reviewer 2 Report

The authors addressed all the raised concern by performing new and additional experiments. This manuscript is now ready for publication.  

Author Response

We would like to thank you for your time and consideration.

Round 3

Reviewer 1 Report

After two rounds of review, the author has made significant improvements to the manuscript.